

# Solid phase extraction and metabolic profiling of exudates from living copepods

Erik Selander[1,2], Jan Heuschele[2], Göran M. Nylund[1], Georg Pohnert[3], Henrik Pavia[1], Oda Bjærke[4], Larisa A. Pender-Healy[5], Peter Tiselius[6] and Thomas Kiørboe[2]

[1] Department of Marine Sciences, University of Gothenburg, Göteborg, Sweden
[2] Centre for Ocean Life, National Institute of Aquatic Resources, Technical University of Denmark, Copenhagen, Denmark
[3] Institute for Inorganic and Analytical Chemistry, Friedrich Schiller University, Jena, Germany
[4] Department of Biosciences, University of Oslo, Oslo, Norway
[5] School of Biology, Georgia Institute of Technology, Atlanta, GA, USA
[6] Department of Biological and Environmental Sciences, University of Gothenburg, Göteborg, Sweden

Corresponding author
Erik Selander,
erik.selander@marine.gu.se

## ABSTRACT

Copepods are ubiquitous in aquatic habitats. They exude bioactive compounds that mediate mate finding or induce defensive traits in prey organisms. However, little is known about the chemical nature of the copepod exometabolome that contributes to the chemical landscape in pelagic habitats. Here we describe the development of a closed loop solid phase extraction setup that allows for extraction of exuded metabolites from live copepods. We captured exudates from male and female *Temora longicornis* and analyzed the content with high resolution LC-MS. Chemometric methods revealed 87 compounds that constitute a specific chemical pattern either qualitatively or quantitatively indicating copepod presence. The majority of the compounds were present in both female and male exudates, but nine compounds were mainly or exclusively present in female exudates and hence potential pheromone candidates. Copepodamide G, known to induce defensive responses in phytoplankton, was among the ten compounds of highest relative abundance in both male and female extracts. The presence of copepodamide G shows that the method can be used to capture and analyze chemical signals from living source organisms. We conclude that solid phase extraction in combination with metabolic profiling of exudates is a useful tool to develop our understanding of the chemical interplay between pelagic organisms.

## INTRODUCTION

Copepods are the most abundant mesozooplankton in the oceans and constitute a central link between primary production and higher trophic levels in the pelagic food web (*Mauchline et al., 1998*; *Turner, 2004*). In addition, copepods change the chemical composition of the surrounding water by taking up (mainly oxygen) and giving off chemical compounds. Common excretion products such as carbon dioxide, ammonium, urea, and dissolved organic carbon (DOC) released during "sloppy feeding" (*Møller, Thor & Nielsen, 2003*) dominate the chemical signature, but copepods also produce

species- and gender-specific compounds that act as inter- and intraspecific signals. Female copepods attract males using sex pheromones. This was first recognized by *Parker (1902)* who observed the behavior of males *Labidocerae* outside caged females that could not be seen by the males while chemical cues could still exit the cage through cotton plugs. This line of research was not pursued further until *Katona (1973)* described the mate finding behavior in *Eurytemora* and *Pseudodiaptomus*. Since then many more examples of chemically mediated mate finding in copepods have been reported (reviewed in *Heuschele & Selander, 2014*). Mate finding involves near field tracking of chemical trails and general search behavior triggered by background levels of female exudates (*Doall et al., 1998*; *Heuschele & Kiørboe, 2012*). Harpacticoids of the genus *Tigriopus* use both diffusible cues and contact chemo-reception to find and identify conspecific females (*Kelly & Snell, 1998*; *Lazzaretto, Salvato & Libertini, 1990*). Sex pheromones increase mate encounter rates several fold compared to random encounters (*Kiørboe, 2008*).

Prey organisms sense copepod exudates and respond by expressing defensive traits. Colonial phytoplankton adjust colony size to evade predation (*Bergkvist et al., 2012*; *Long et al., 2007*), and harmful algal bloom-forming dinoflagellates increase toxin content (*Selander et al., 2006*; *Wohlrab, Iversen & John, 2010*) or change swimming behavior to avoid copepod encounters (*Jiang, Lonsdale & Gobler, 2010*; *Selander et al., 2011*). Phytoplankton exposed to copepod exudates becomes more resistant to grazing and harmful metabolites may affect a wide variety of organisms in the pelagic food web (*Bergkvist et al., 2012*; *Long et al., 2007*; *Selander et al., 2006*). Copepod exudates consequently modulate pelagic food webs in many ways, and beyond the direct consumption of prey.

Despite the many documented effects of copepod exudate our understanding of the chemical composition is mainly limited to un-characterized dissolved organic matter and nitrogen excretion products (*Breckels et al., 2013*; *Perez-Aragon, Fernandez & Escribano, 2011*; *Saba, Steinberg & Bronk, 2011*). A group of eight taurine conjugated lipids that induce toxin formation in dinoflagellates, copepodamides, are hereto the only identified info-chemicals from copepod sources (*Selander et al., 2015*). Metabolic profiling provides an opportunity to characterize exudates from copepods in greater detail. In mass spectrometric metabolomic profiling the mass of all detectable compounds is recorded with high accuracy providing a list of exuded compounds of unknown identity but with a known molecular mass. Samples from different organisms or sexes can subsequently be cross referenced to identify compounds that are associated with an organism or a specific biological activity. Candidate compounds can then be tested for biological activity and active compounds structurally determined. As an example, the first diatom pheromone, diproline, was recently identified by comparing the metabolic profiles of exudates from sexual and asexual cultures of the diatom *Seminavis robusta* (*Gillard et al., 2013*). Diproline was found in higher concentrations in the sexual culture, and subsequently proven active in bioassays. The species examined here, *Temora longicornis*, has been suggested to produce sex pheromones based on the observation that males track the chemical wake of females (*Doall et al., 1998*). Males, however, sometimes track their own wakes, which suggest that the signal compound(s) guiding males may not be entirely sex specific. In fact, *Temora longicornis* males even pursue females of the partially sympatric species *Temora stylifera*

showing that the signals to some extent overlap between species (*Goetze & Kiørboe, 2008*). Furthermore, *Temora longicornis* exude compound(s) that suppress chain formation in diatoms which allows the smaller algal units to escape grazers (*Bergkvist et al., 2012*). Together these observations make *Temora longicornis* an interesting target for exudate profiling.

We designed a sampling devise to capture and analyze exudates from live male and female *Temora longicornis*. The main objective was to obtain more detailed information about the copepod exometabolome. We specifically looked for differences in exudate composition between the two sexes to test the hypothesis that the sexual dimorphism in copepod morphology and behavior (*Gilbert & Williamson, 1983*) also manifest in terms of a gender specific exudate profile.

## MATERIALS AND METHODS

### Sampling and sorting of copepods

Copepods were collected with oblique tows from the surface to ∼20 m depth in the Kosterfjord outside Sven Lovén Centre for Marine Sciences, Tjärnö on the west coast of Sweden in January 2013. A 200-µm mesh size WP-2 net equipped with a non-filtering cod end was used to reduce physical stress. The catch was directly transferred to the laboratory. Adult males and females were pipetted to separate jars under dissecting microscopes. The copepods were fed a mixture of *Rhodomonas* sp., *Chaetoceros* sp., and *Skeletonema* sp. before and between trials. Algae were cultured in f/2 medium in a light and temperature controlled room set at 16 °C and 16:8 h light:dark cycles. Copepods were picked continuously during the experimental period, and had been kept in captivity for 1–5 days before use in extractions.

### Closed loop extraction of copepod exudates

To obtain a low and constant background of dissolved organic compounds in the incubation water we prepared a batch of purified seawater (pSW) that was used in all incubations. The water (33 psu) was filtered through GF/F filter (Whatman, Little Chalfont, UK) and pumped (∼7 ml min$^{-1}$) through two serial isolute ENV+ (Biotage, Uppsala, Sweden) 500 mg solid phase extraction (SPE) columns. The SPE purification reduced the background of retainable DOC and increased the signal to noise ratio in copepod incubations considerably. The pSW was stored refrigerated (∼5 °C) in the dark.

Animals for each replicate extraction were given 30–45 min to clear their alimentary tract in 50-mL polypropylene cylinders with a 200-µm nylon mesh bottom standing in glass beakers with pSW. This was done to decrease the contribution of compounds associated with fecal pellets in the samples. The mesh bottom cylinders were subsequently sequentially dipped in two freshly prepared glass beakers with pSW to eliminate carry over from culture water, feed algae, or fecal pellets. After the final rinse, the copepods were gently flushed into an Erlenmeyer flask in 150 ml pSW. Exudates were stripped from the incubation water using a closed loop solid phase extraction system (Fig. 1). The number of animals used per incubation ranged from 239 to 619, with an average of 376 individuals. The high density of copepods was necessary to obtain sufficient signal to noise ratio. Water was
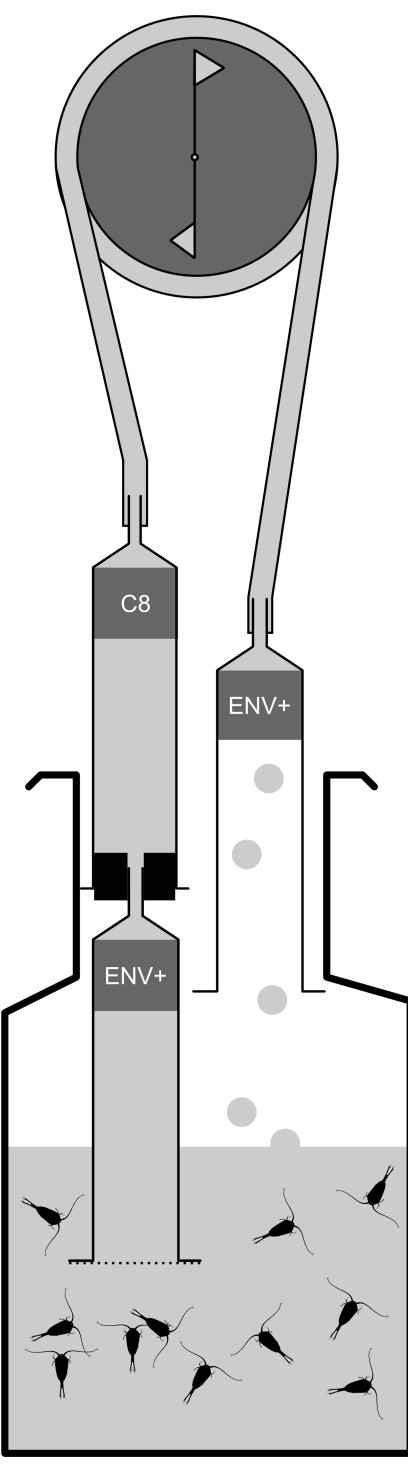

**Figure 1  Sampling device used to collect exudates from copepods.** Water is drawn through the partially immersed ENV+ and the following C8 SPE before reintroduced into the incubation chamber. The ENV+ column on the outlet end serves to minimize contamination from pump or hosing. The hatched line in the opening of the first SPE column represents a 30 μm nylon plankton mesh that prevents copepods and copepod eggs from entering the column.

drawn through an ENV+ (100 mg; Biotage, Uppsala, Sweden) solid phase extraction (SPE) column followed by a serially connected 100 mg C8 SPE (Biotage, Uppsala, Sweden) by a peristaltic pump ($\sim$1.5 ml min$^{-1}$). The ENV+ is a polystyrene resin functionalized with phenolic groups. It does not collapse in water and retains more polar compounds compared to the C8 column. The combination of ENV+ and traditional silica based reversed phase (RP) packing has been successfully used to retain info-chemicals from parasitic copepods (*Ingvarsdottir et al., 2002*) and functionalized polystyrene resins generally retain a wider range of DOC from natural seawater samples compared to silica based RP resins (*Dittmar et al., 2008*). Columns were activated with 3 ml methanol followed by 5 ml distilled water prior to use. The water from the pump was reintroduced into the incubation chamber through a 100 mg ENV+ column to minimize contamination from the pump or tubing (Fig. 1). The inlet to the serial columns was covered by a 30 $\mu$m nylon plankton mesh to avoid copepods and eggs from entering the columns. Control samples were obtained as above, but without copepods in the incubations. We obtained five replicate extractions from each treatment (male, female, and control).

After an overnight 12 h incubation period the return line was removed and sampling continued until $\sim$50 ml was left in the flask. The duration of the incubation was noted, and the columns removed. The columns were desalted with 1 mL ultrapure water (Millipore) and eluted with 1.5 ml 100% LC-MS grade methanol (Lichrosolv; Merck, Kenilworth, NJ, USA) with a 30 s soak step after 0.75 ml to increase yield. The combined eluate (from the ENV+ and C8 SPE) was evaporated under a stream of nitrogen at 30 °C. The sample was resolved in 150 $\mu$l methanol and stored in sealed HPLC vials with inserts at $-20$ °C until use in bioassays and analysis.

The copepods from each incubation were photographed (Canon 7D equipped with a 65 mm macro lens; Sigma, Kanagawa, Japan) in a Petri dish on a light table for enumeration before transfer to larger volume beakers and supplemented with food. Used animals were allowed to feed during the day and were reused the following night. Freshly picked and reused animals were used in separate incubations. Thus, all incubations except the first one consisted of a replicate with freshly isolated copepods, and one with mainly reused copepods. We could not see any systematic difference between fresh and reused copepods in the exudate profiles and treated each individual incubation as a replicate in the multiviariate analysis. Incubations were done in a dark thermostatic room at 10 °C.

## Mass spectrometry

Metabolic profiling samples were analyzed using an Agilent 1100 HPLC equipped with a Licrosphere 2.1 $*$150 mm, 3 $\mu$m C18 silica column (Poroshell; Agilent, Santa Clara, CA, USA) with a Q-Tof 6540 MS as detector. The eluent gradient started at 5% acetonitrile and 0.1% formic acid in water (Eluent A) that was maintained for one minute followed by a linear gradient up to 95% acetonitrile and 0.1% formic acid (Eluent B) over 14 min. 95% Acetonitrile was maintained for 4 min and the column re-equilibrated for 7 min in 100% eluent A before the following injection. Injection volume was 4 $\mu$l and eluent flow rate 250 $\mu$l min$^{-1}$. Positive mass spectra were acquired in 4 GHz High Resolution mode with 2 spectra s$^{-1}$ sampled over a scan range of 80–1,100 *m/z*. ESI conditions were

gas temperature 300 °C, drying gas 8 l min$^{-1}$, nebulizer 40 psi, sheath gas temperature 350 °C, sheath gas flow 11 l min$^{-1}$, nozzle voltage 500 V, fragmentor 120 V, and skimmer 65 V. Inspection of resulted TIC chromatograms showed that two samples (one control and one female) dramatically deviated from replicate samples and were excluded from further analysis.

The same settings were used for tandem MS experiments targeting compound #10 with m/z 434.28 and collision energy set to 40 V.

## Multivariate analysis

Mass spectra were de-convoluted by the molecular feature extraction algorithm (MFE) of the Agilent MassHunter Qualitative Analysis software (MH Qual., version B.05.00). MFE locates groups of co-variant ions, and each group represents a unique feature. The features are defined as time-aligned ions (i.e., isotopes, adducts, dehydrations, and/or dimers) summarized to the calculated neutral mass, possess an abundance and a retention time (RT), and are subsequently treated as compounds (*Sana et al., 2008*). We allowed for H$^+$, Na$^+$, K$^+$ and NH$_4^+$ adducts and a neutral dehydration. Compounds with absolute peak heights of 10,000 or higher were selected and stored as compound exchange format (CEF) files for compound alignment in Mass Profiler Professional (Agilent Technology, MPP, version B.12.01). Alignment parameters were 0.2 min RT window and 5 ppm + 2 mD mass tolerance. To reduce false-positive and false-negative detection rates from the untargeted MFE, we created a composite list of ions generated in MPP for a recursive, targeted analysis of raw data using the 'find by ion' algorithm of MH Qual. The settings for adduct ions, neutral mass, and aggregates (i.e., dimers) were the same as for MFE. The recursive CEF files were re-imported into MPP for alignment using the previous setting. The aligned data were filtered by considering only compounds (i.e., metabolites) occurring in at least 3 replicates within any experimental group, and the resulted aligned compound list was exported as a text file for further statistical analysis.

Multivariate data analysis was performed with the software SIMCA (version 13.0.0.0, Umetrics, Sweden). Principal component analysis (PCA) was used to generate an overview and to analyze for systematic differences in metabolite content. The metabolite data were modeled and interpreted using orthogonal partial least squares with discriminant analysis (*OPLS-DA*) (*Bylesjo et al., 2006*; *Trygg & Wold, 2002*). *OPLS-DA* is a supervised multivariate data projection method used to relate a set of predictor variables ($X$ or metabolites in this study) to a response matrix ($Y$) that represents predefined sample classes (i.e., control, female and male copepods). This method can then be used to predict class identity and to extract specific features among the predictor variables (metabolites) distinguishing between the predefined sample classes. *OPLS* operates by separating the systematic variation in $X$ into two parts: one that is linearly related to $Y$, and thus can be used to predict $Y$, and one that is orthogonal and uncorrelated to $Y$. To determine copepod specific compounds, *OPLS-DA* was carried out between controls and males in one model, and between controls and females in a separate model. Correlation coefficients obtained from the *OPLS-DA* models were used to identify metabolites significantly more abundant in males and/or females, using correlation coefficients greater or equal to 0.75 as a cut-off.

The chromatographic peaks of the metabolites were visually inspected for each sample, and noise peaks (spikes and/or lack of regular peak shape and symmetry) together with peaks with high intensity in copepod free controls were removed. The resulting metabolite data set, normalized to the numbers of incubated copepods, was further analyzed with both *PCA* and *OPLS-DA* for systematic differences between male and female copepods. Statistically significant metabolites related to differences between male and females were selected from both loadings plots showing *OPLS-DA*-derived correlation (Fig. 4A) and covariance (Fig. 4B) coefficients, the latter with jack-knifed 95% confidence intervals calculated from cross validation (*Wiklund et al., 2008*). The correlation loading profile relates to the effect and reliability of each metabolite for class separation, and covariance loading profile relates to the contribution or magnitude of each metabolite. Thus, the combination of covariance and correlation information is an effective method of finding metabolites of interest (*Wiklund et al., 2008*).

The quality of the *OPLS-DA* models was assessed by the parameters $R^2X$ and $Q^2$, which represent the total explained variations for $X$ matrix and the model predictability, respectively (*Bylesjo et al., 2006*; *Wiklund et al., 2008*). The models were certified using a 7-fold cross-validation method and a permutation test, using the default option of SIMCA for cross-validation. A model was considered significant if the $Q^2$ value was significant ($P < 0.05$) through permutation. The data sets were Pareto-scaled prior to all *PCA* and *OPLS-DA*. For calculations of elemental compositions of specific metabolites, the molecular formula generator (MFG) algorithm included in the Agilent MassHunter Workstation software was used. MFG uses a range of MS information including accurate mass measurements, isotope abundances, and spacing between isotope peaks to produce a list of candidate molecular formulas that are ranked according to their relative probabilities (*Sana et al., 2008*).

## Bioassay of exudate samples

To bioassay female exudate samples for sex pheromone activity, 50 µl of the combined female exudates were dried down and resolved in 100 µl pSW with slightly elevated (34 psu compared to the 33 psu of incubation water) salinity. 50 µl of this solution was added to a 100 µl pipet tip ending with a 0.25 mm ø PEEK capillary to restrict flow. The pipette tip was allowed to empty due to gravitation into a NUNC cell culture bottle (inner dimensions: $65 \times 38 \times 22$ mm) with 55 ml seawater and 50 inexperienced male copepods. The setting was kept in the dark at 10 °C and illuminated from below with infrared light. The formation of stabile scented trails was verified in test trials with fluorescent dye (fluorescin). Each replicate was filmed for 10 min with a camera (Sony DCR-TRV738E) directed towards the side of the aquaria (perpendicular to the light path of the infrared light providing dark field visualization of copepods) and analyzed for the occurrence of trail following behavior, i.e., the occurrence of fast directional swimming patterns coinciding with the location of the chemical trail. It is likely that only a fraction of the females were emitting pheromones, and that starvation and poor recoveries of compounds through sample preparation may have lowered the concentration further. To compensate for this, the amount added over the 10 min corresponded to the exudates from 125 females over 12 h.

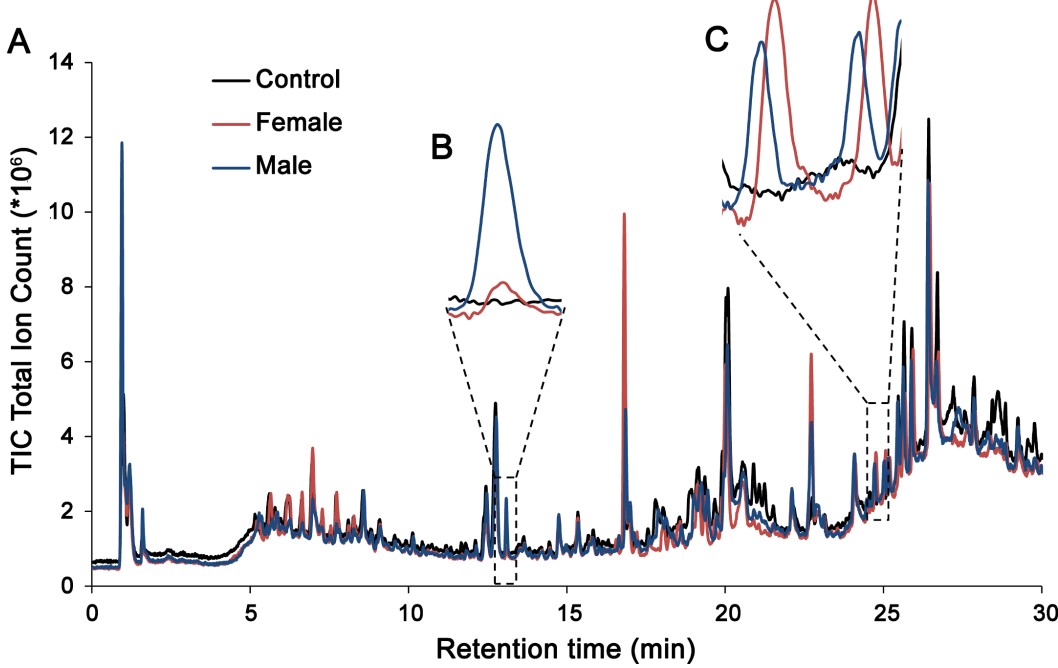

**Figure 2** (A) Representative chromatograms from male exudates (blue), female exudates (red), and copepod free controls (black). Prominent peaks exclusive to copepod samples were always visible after 13 min (compound 10 in Table 2 with a monoisotopic mass of 433.29 Da), enlarged in (B)) as well as at 24–25 min (compounds 58 and 63 in Table 1, monoisotopic mass 563.39 and 640.41 Da, enlarged in (C)). The majority of the 87 compounds detected were, however, too low in concentration to be observed without extracting specific masses.

# RESULTS

## Chemical analysis

The analyses revealed a high complexity of compounds with signals observed over the entire polarity range of the LC-gradient. With a few exceptions, the copepod-derived compounds were present in amounts not directly visible in the TIC chromatogram (Fig. 2). Despite the efforts to purify the medium, several signals were observed in the copepod-free controls, and these were excluded from further data evaluation as described below. Raw data in the form of an aligned peak list is available in Supplemental Information 1.

A *PCA* representation showed a clear separation between copepod samples and controls (Fig. 3A). The first two principal components explain 21.3 and 19.7% of the total variance, respectively. Male and female exudates samples cluster together showing that differences between the sexes cannot be identified in this untreated data-set. Orthogonal partial least square discriminant analysis (*OPLS-DA*) showed a clear separation between control and male samples, as well as between control and female samples (Table 1). The separation could be attributed to 115 metabolites significantly more abundant in male and/or female copepod exudate samples compared to controls (correlation coefficients ≥0.75), and removal of noise peaks and peaks that were also present in relatively high intensity in copepod-free controls decreased the number to 87. *PCA* on this reduced metabolite data

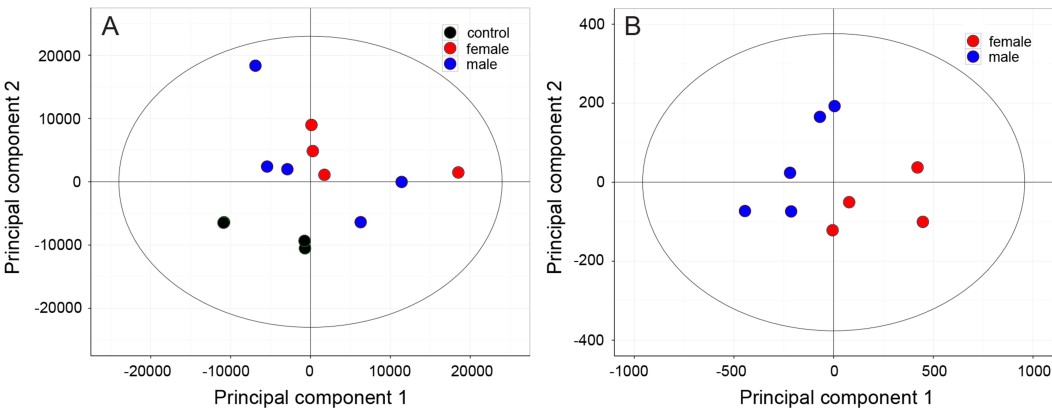

**Figure 3** (A) PCA based on Pareto-scaled raw data showing the separation between sea water controls (black), male (red) and female (blue) exudate samples. (B) PCA on the reduced data-set containing only compounds indicated to be of copepod origin in the control-male and control-female OPLS-DA analysis show a systematic difference also between male and female exudates.

**Table 1** Parameter values of OPLS-DA models.

| OPLS-DA model | No[a] | R2X(cum)[b] | R2X[c] | Q2(cum)[d] |
|---|---|---|---|---|
| Control vs. male | 1P + 2O | 0.537 | 0.213 | 0.785 |
| Control vs. female | 1P + 2O | 0.631 | 0.292 | 0.823 |
| Male vs. female | 1P + 2O | 0.874 | 0.465 | 0.87 |

**Notes.**
[a] No, the number of predictive (P) and orthogonal (O) components.
[b] The R2X(cum) value is the predictive and orthogonal variation in model samples ($X$) explained by the model.
[c] The R2X value is the amount of variation in $X$ which is correlated to $Y$ (response matrix).
[d] The Q2(cum) value describes the predictive ability of the model, based on sevenfold-cross validation.

set revealed differences also between male and female copepod samples (Fig. 3B), with 71.0 and 10.9% of the variance explained by the two principal components. To identify the metabolites responsible for the differentiation between female and male samples, loading plots with *OPLS-DA*-derived correlation and covariance coefficients with jack-knifed 95% confidence intervals were generated (Figs. 4A, 4B and Table 1). The resulting plots showed that exudation of metabolites normalized to the number of copepods were higher for females compared to males (Fig. 4B, positive values indicates more in female than male samples). The difference was mainly quantitative and only two metabolites were unique for female exudates. A single metabolite had higher abundance in male exudates, suggested by negative correlation and covariance coefficients with jack-knifed confidence interval not overlapping zero. That metabolite, however, had a relatively low correlation coefficient. Most of the metabolites were only slightly more abundant in female samples indicated by their low covariance, but seven metabolites were distinct in that they had high covariance and correlation (Figs. 4A and 4B).

A complete compound list showing all 87 metabolites considered to be of copepod origin is given in Table 2. The seven compounds most strongly associated with female copepods are highlighted together with the two compounds exclusively found in female samples. As male *Temor* as are known to track female wakes (*Doall et al., 1998*) we tested the exudates

**Table 2  Metabolites from female and male *Temora longicornis*.** The two compounds exclusively found in female exudates are indicated in red, and metabolites that are significantly more abundant in females but also present in male exudates in grey.

| No. | RT (min) | Mass[a] | Fold change[b] | Elemental composition with scores[c] |
|---|---|---|---|---|
| 1 | 1.59 | 136.0382 | 0.5 | $C_5H_4N_4O$ (98.0), $C_4H_8O_5$ (93.6) |
| 2 | 1.70 | 152.0331 | 0.4 | $C_5H_4N_4O_2$ (98.0), $C_4H_8O_6$ (92.4) |
| 3 | 4.86 | 284.1373 | 0.9 | $C_{13}H_{20}N_2O_5$ (99.6), $C_{14}H_{16}N_6O$ (93.2), $C_8H_{17}N_{10}P$ (91.8) |
| 4 | 6.17 | 304.1495 | 7.4 | $C_{10}H_{20}N_6O_5$ (98.9), $C_{11}H_{16}N_{10}O$ (91.8), $C_{12}H_{25}N_4OPS$ (91.0) |
| 5 | 6.75 | 264.1473 | 1.6 | $C_{14}H_{20}N_2O_3$ (98.8) |
| 6 | 9.34 | 163.0993 | 0.8 | $C_{10}H_{13}NO$ (98.8) |
| 7 | 10.61 | 230.1514 | 1.6 | $C_{12}H_{22}O_4$ (99.0) |
| 8 | 11.88 | 334.2144 | 0.3 | $C_{20}H_{30}O_4$ (99.2), $C_{21}H_{26}N_4$ (92.4), $C_{14}H_{31}N_4O_3P$ (90.5) |
| 9 | 12.76 | 380.2533 | 1.6 | N/A |
| 10 | 13.08 | 433.2863 | 0.9 | $C_{22}H_{43}NO_5S$ (99.1), $C_{23}H_{39}N_5OS$ (95.2), $C_{17}H_{40}N_9PS$ (95.1) |
| 11 | 13.08 | 505.2166 | 0.9 | $C_{23}H_{39}NO_7S_2$ (94.6), $C_{24}H_{43}NO2S_4$ (92.9), $C_{27}H_{41}NP_2S_2$ (90.1) |
| 12 | 14.15 | 765.4143 | 1.3 | $C_{36}H_{63}NO_{16}$ (99.5), $C_{31}H_{60}N_9O_{11}P$ (98.5), $C_{34}H_{51}N_{15}O_6$ (98.4) |
| 13 | 14.23 | 362.2431 | 1.7 | $C_{18}H_{30}N_6O_2$ (95.5), $C_{19}H_{41}P_3$ (93.9) |
| 14 | 14.97 | 364.2599 | 1.2 | $C_{16}H_{37}N_4O_3P$ (97.8), $C_{18}H_{32}N_6O_2$ (93.7), $C_{22}H_{36}O_4$ (90.1) |
| 15 | 14.98 | 661.1729 | 1.2 | $C_{25}H_{41}ClNO_{13}PS$ (96.1), $C_{30}H_{47}ClNOP_3S_3$ (95.7), $C_{20}H_{36}O_3$ (98.3) |
| 16 | 15.20 | 364.2592 | 1.3 | $C_{18}H_{32}N_6O_2$ (98.3) |
| 17 | 15.20 | 364.2592 | 1.4 | $C_{18}H_{32}N_6O_2$ (98.3), $C_{19}H_{43}P_3$ (94.3), $C_{16}H_{37}N_4O_3P$ (92.7) |
| 18 | 15.73 | 364.2589 | 1.4 | $C_{18}H_{32}N_6O_2$ (97.2), $C_{19}H_{43}P_3$ (94.4) |
| 19 | 16.82 | 174.1042 | 1.7 | N/A |
| 20 | 16.82 | 602.3516 | 2.1 | $C_{33}H_{46}N_8OS$ (93.0) |
| 21 | 16.82 | 367.1243 | 1.5 | N/A |
| 22 | 16.82 | 226.1720 | 1.7 | $C_{17}H_{22}$ (99.0) |
| 23 | 16.82 | 176.1194 | 1.6 | N/A |
| 24 | 16.82 | 312.1701 | 1.9 | $C_{16}H_{20}N_6O$ (99.1), $C_{16}H_{28}N_2S_2$ (96.6), $C_{13}H_{29}O_6P$ (95.3) |
| 25 | 16.82 | 130.0778 | 1.7 | N/A |
| 26 | 16.82 | 272.1775 | 1.8 | $C_{18}H_{24}O_2$ (99.6), $C_{12}H_{25}N_4OP$ (91.3) |
| 27 | 16.82 | 134.0725 | 1.6 | N/A |
| 28 | 16.82 | 634.2966 | 2.2 | $C_{38}H_{50}O_2S_3$ (96.5), $C_{32}H_{51}N_4OPS_3$ (95.8), $C_{30}H_{46}N_6O_5S_2$ (94.0) |
| 29 | 16.82 | 132.0943 | 1.7 | N/A |
| 30 | 16.82 | 344.1074 | 2.1 | $C_{19}H_{20}O_4S$ (93.2) |
| 31 | 16.82 | 618.3222 | 1.8 | $C_{36}H_{47}N_2O_5P$ (97.9), $C_{34}H_{50}O_8S$ (96.6), $C_{38}H_{52}OP_2S$ (96) |
| 32 | 16.82 | 118.0778 | 1.1 | $C_9H_{10}$ (92.7) |
| 33 | 16.82 | 158.1093 | 1.6 | N/A |
| 34 | 16.82 | 908.5126 | 20.9 | $C_{52}H_{78}O_9P_2$ (99.4), $C_{53}H_{74}N_4O_5P_2$ (98.7), $C_{46}H_{64}N_{14}O_6$ (98.6) |
| 35 | 16.82 | 144.0935 | 1.7 | N/A |
| 36 | 16.82 | 146.1091 | 1.7 | N/A |
| 37 | 16.82 | 186.1402 | 1.6 | N/A |
| 38 | 16.82 | 254.1667 | 1.6 | $C_{18}H_{22}O$ (99.1) |
| 39 | 16.83 | 328.1416 | 1.4 | $C_{20}H_{25}PS$ (95.4) |
| 40 | 16.83 | 92.0621 | 1.4 | N/A |
| 41 | 17.21 | 274.1937 | 1.6 | $C_{18}H_{26}O_2$ (98.7) |
| 42 | 17.73 | 300.2092 | 1.0 | $C_{20}H_{28}O_2$ (97.2) |
| 43 | 18.05 | 610.2795 | 0.8 | $C_{35}H_{38}N_4O_6$ (99.5), $C_{37}H_{45}N_2P_3$ (98.7), $C_{36}H_{34}N_8O_2$ (97.8) |
| 44 | 18.31 | 300.2085 | 1.1 | N/A |
| 45 | 18.79 | 326.2249 | 0.7 | $C_{22}H_{30}O_2$ (94.9) |
| 46 | 19.00 | 435.3027 | 1.1 | $C_{23}H_{41}N_5OS$ (94.0), $C_{24}H_{42}N_3O_2P$ (90.4), $C_{22}H_{45}NO_5S$ (90.4) |
| 47 | 20.80 | 274.1943 | 1.7 | $C_{18}H_{26}O_2$ (92.0) |
| 48 | 21.51 | 552.3253 | 1.4 | $C_{38}H_{40}N_4$ (98.5), $C_{37}H_{44}O_4$ (96.8), $C_{30}H_{46}N_6P_2$ (96.4) |
| 49 | 21.63 | 622.4023 | 0.7 | $C_{42}H_{54}O_4$ (99.6), $C_{43}H_{50}N_4$ (96.7), $C_{36}H_{55}N_4O_3P$ (95.1) |
| 50 | 21.99 | 598.4036 | 0.6 | $C_{41}H_{50}N_4$ (98.6), $C_{40}H_{54}O_4$ (97.0), $C_{33}H_{56}N_6P_2$ (96.0) |

**Table 2** (*continued*)

| No. | RT (min) | Mass[a] | Fold change[b] | Elemental composition with scores[c] |
|---|---|---|---|---|
| 51 | 22.70 | 534.2645 | 2.4 | $C_{31}H_{39}N_2O_4P$ (99.8), $C_{33}H_{34}N_4O_3$ (95.3), $C_{32}H_{35}N_6P$ (95.1) |
| 52 | 23.11 | 595.1772 | 1.7 | $C_{36}H_{26}ClN_5O_2$ (93.1), $C_{36}H_{34}ClNOS_2$ (92.2), $C_{29}H_{26}ClN_{11}S$ (91.5) |
| 53 | 23.54 | 534.3936 | 2.0 | $C_{33}H_{50}N_4O_2$ (94.8), $C_{32}H_{54}O_6$ (91.4), $C_{31}H_{55}N_2O_3P$ (90.5) |
| 54 | 24.07 | 560.4090 | 1.6 | $C_{35}H_{52}N_4O_2$ (99.4), $C_{34}H_{56}O_6$ (96.1), $C_{33}H_{57}N_2O_3P$ (93.3) |
| 55 | 24.21 | 596.3877 | 1.6 | $C_{41}H_{48}N_4$ (98.8), $C_{40}H_{52}O_4$ (97.4), $C_{33}H_{54}N_6P_2$ (94.3) |
| 56 | 24.48 | 581.3995 | 1.2 | N/A |
| 57 | 24.47 | 596.4063 | N/A | $C_{34}H_{48}N_{10}$ (98.3), $C_{31}H_{57}N_4O_5P$ (97.6), $C_{33}H_{52}N_6O_4$ (96.5) |
| 58 | 24.73 | 563.3923 | 1.5 | N/A |
| 59 | 24.73 | 546.3871 | N/A | $C_{40}H_{50}O$ (92.0) |
| 60 | 24.82 | 652.2914 | 1.4 | $C_{38}H_{36}N_8O_3$ (98.5), $C_{37}H_{52}O_2P_4$ (97.1), $C_{35}H_{45}N_2O_8P$ (96.8) |
| 61 | 24.83 | 581.4016 | 1.2 | $C_{43}H_{51}N$ (91.2) |
| 62 | 25.02 | 565.4044 | 1.5 | $C_{38}H_{51}N_3O$ (95.1), $C_{36}H_{56}NO_2P$ (95) |
| 63 | 25.04 | 640.4123 | 0.7 | $C_{36}H_{57}N_4O_4P$ (99.1), $C_{37}H_{53}N_8P$ (97.4), $C_{34}H_{62}N_2O_5P_2$ (97.4) |
| 64 | 25.16 | 565.4049 | 2.1 | $C_{36}H_{56}NO_2P$ (98.7), $C_{38}H_{51}N_3O$ (94.9), $C_{30}H_{57}N_5OP_2$ (90.4) |
| 65 | 25.16 | 548.4030 | 2.9 | $C_{40}H_{52}O$ (96.8), $C_{32}H_{58}N_2OP_2$ (90.5) |
| 66 | 25.37 | 588.4407 | 1.6 | $C_{37}H_{56}N_4O_2$ (98.2), $C_{35}H_{61}N_2O_3P$ (93.6), $C_{36}H_{60}O_6$ (92.6) |
| 67 | 25.59 | 639.4035 | 0.4 | N/A |
| 68 | 25.64 | 563.3889 | 1.9 | $C_{36}H_{54}NO_2P$ (98.5), $C_{38}H_{49}N_3O$ (95.9) |
| 69 | 25.72 | 639.4068 | 0.5 | N/A |
| 70 | 25.88 | 790.4619 | 1.4 | N/A |
| 71 | 25.89 | 785.5103 | 1.5 | $C_{47}H_{63}N_9O_2$ (98.5), $C_{48}H_{74}N_3P_3$ (98.0), $C_{44}H_{72}N_3O_7P$ (97.9) |
| 72 | 26.09 | 981.6413 | 2.0 | $C_{52}H_{92}N_3O_{12}P$ (99.4), $C_{47}H_{89}N_{11}O_7P_2$ (98.8), $C_{48}H_{100}N_5O_5P_5$ (98.6) |
| 73 | 26.09 | 986.5968 | 2.0 | $C_{53}H_{74}N_{14}O_5$ (99.0), $C_{60}H_{84}N_4O_4P_2$ (98.7), $C_{55}H_{86}O_{15}$ (97.7) |
| 74 | 26.20 | 787.5238 | 1.8 | $C_{45}H_{73}NO_{10}$ (97.7), $C_{40}H_{70}N_9O_5P$ (96.1), $C_{40}H_{80}N_5O_2P_3S$ (95.7) |
| 75 | 26.61 | 366.3292 | 0.9 | $C_{27}H_{42}$ (97.7) |
| 76 | 26.72 | 605.4007 | 1.5 | $C_{45}H_{51}N$ (93.6) |
| 77 | 26.75 | 835.5826 | 1.8 | $C_{58}H_{78}NOP$ (98.5), $C_{50}H_{84}N_3OP_3$ (98.3), $C_{49}H_{73}N_9O_3$ (97.8) |
| 78 | 27.43 | 739.5623 | 1.8 | $C_{43}H_{83}NO_2P_2S$ (99), $C_{36}H_{73}N_{11}O_3S$ (96.4), $C_{39}H_{81}NO_9S$ (96.2) |
| 79 | 27.74 | 384.3400 | 1.4 | $C_{27}H_{44}O$ (93.0) |
| 80 | 27.99 | 549.4112 | 2.5 | N/A |
| 81 | 28.01 | 652.5047 | 1.9 | $C_{39}H_{75}OP_3$ (96.0), $C_{36}H_{69}N_4O_4P$ (95.3), $C_{35}H_{73}O_8P$ (94.3) |
| 82 | 28.35 | 565.4058 | 2.1 | N/A |
| 83 | 28.38 | 549.4112 | 2.3 | $C_{36}H_{56}NOP$ (95.7), $C_{35}H_{55}N_3S$ (90.2) |
| 84 | 28.75 | 565.4719 | 1.9 | $C_{35}H_{59}N_5O$ (93.4), $C_{33}H_{64}N_3O_2P$ (92.8), $C_{34}H_{63}NO_5$ (91.8) |
| 85 | 29.03 | 428.3664 | 1.6 | $C_{29}H_{48}O_2$ (95.3) |
| 86 | 29.86 | 916.5733 | 2.2 | $C_{54}H_{81}N_2O_8P$ (98.6), $C_{49}H_{88}N_6P_4S$ (97.9), $C_{52}H_{84}O_{11}S$ (97.4) |
| 87 | 29.86 | 937.5355 | 2.1 | N/A |

**Notes.**
[a] Calculated neutral mass.
[b] Fold change female/male. Metabolite 57 and 59 were not detectable for males.
[c] The three highest ranked candidate molecular formulas. Only candidates with scores ≥90 were included.

in a bioassay monitoring mate finding behavior in males exposed to an artificial trail tainted with the female exudates. No mate tracking behaviors were, however, observed in the trail following bioassays of the combined ENV+ and C8 SPE extracts.

The most evident copepod compound in the TIC chromatogram, compound 10 (Fig. 1 and Table 2) with high relative abundance in both male and female exudates had an accurate mass and retention time (13.1 min) matching that of dehydrated copepodamide G (Table 2 compound 10, m/z 433.2865, [M-H2O], Δ0.7 ppm). Similar to copepodamide G compound 10 also showed an additional dehydration (m/z 416.2833, [M-2H2O + H]$^+$, Δ0.5 ppm) in positive mode. MS/MS experiments further revealed a fragment of

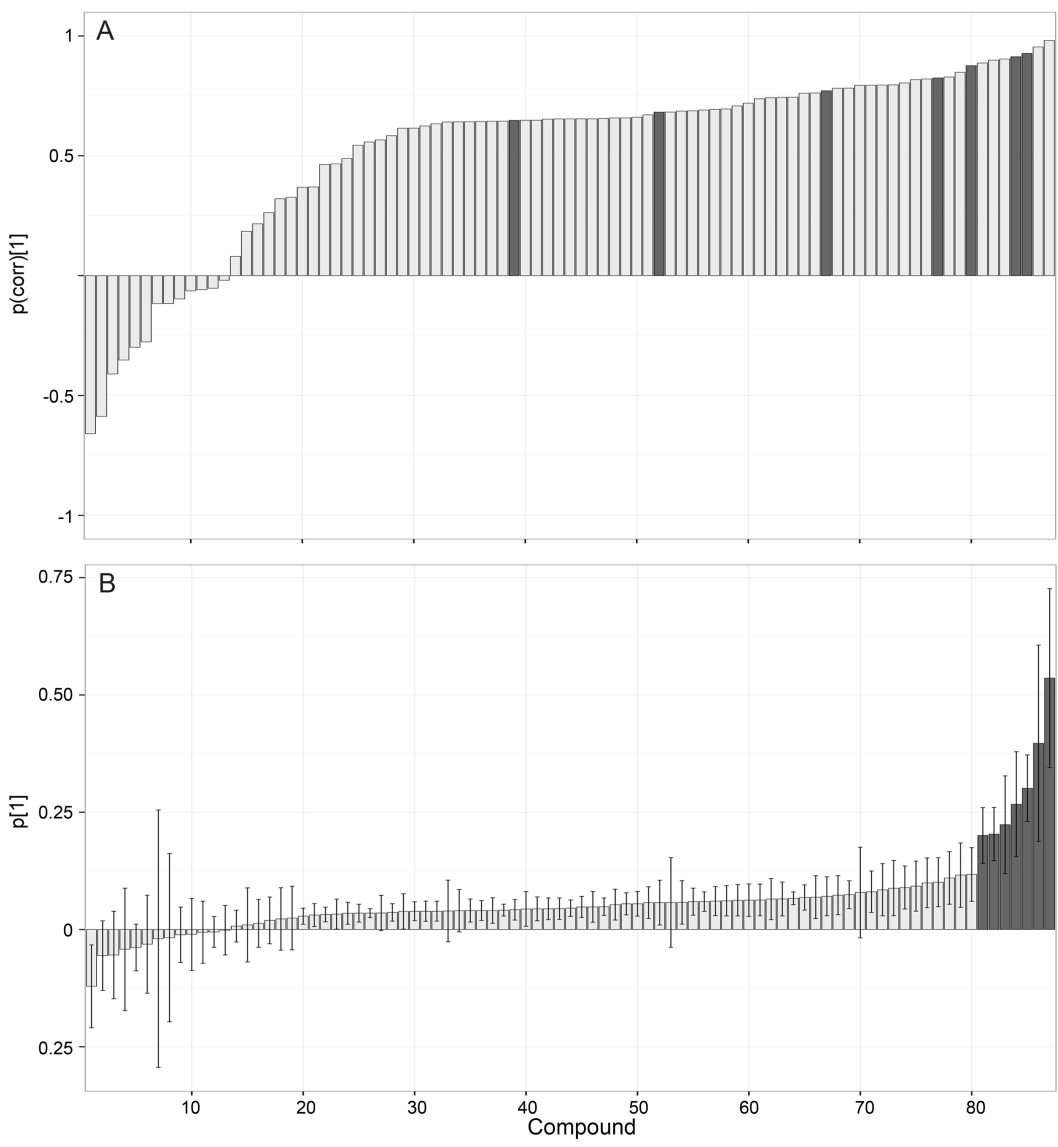

**Figure 4 Differences between male and female exudates compound by compound.** The OPLS-DA loading plots of metabolites exuded from female and male copepods. The loading plot (A) represents the effect and reliability (correlation, $p(corr)[1]$), and (B) the contribution or magnitude (covariance, $p[1]$) of each metabolite to class separation. Jack-knifed confidence intervals (95%) in the covariance plot (B) were calculated from cross-validation. Metabolites with high $p(corr)[1]$ and $p[1]$ values are shaded.

m/z 126.0219, indicating the presence of taurine ($C_2H_8NO_3S$, $\Delta 4.7$ ppm) also present in Copepodamide G.

## DISCUSSION

Copepods release only minute amounts of metabolites into their environment. To overcome the limitations associated with the analytics of such dilute cues we developed a closed loop extraction technique with serial solid phase extraction columns to cover a wide polarity range.

Metabolic profiling did provide more comprehensive information about the copepod exometabolome than has previously been available. The approach is especially useful for small and manually sorted organisms as mass spectrometry is a sensitive technique that requires little material. Yet, the method has some significant constraints. Only compounds that are retained and detectable will be included in the analysis. The most polar metabolites and proteins would for example not be recovered on the column resins used here. Highly volatile compounds would also have been lost during the evaporation of the eluates. By using different types of columns resins, more targeted extractions can be envisioned that could be paired with adapted LC separation techniques. Sequestering the more polar analytes in sea water is a challenge that can be addressed, e.g., by extraction and separation using hydrophilic/lipophilic balanced interaction chromatography (*Spielmeyer & Pohnert, 2010*). Anion and cation exchange may also be useful, especially in fresh water organisms, as the high salt concentration in seawater will compromise its function. Collection over several hours enables high extraction success and covers metabolites that are released during brief periods. At the same time, sampling over long time periods may increase the risk of degradation and confounding effects of starvation. Analyte sequestering time is consequently a tradeoff between quantity and quality that will have to be carefully considered for each individual case.

Despite the fact that we pre-purify instrumentation and depleted seawater in organic constituents by solid phase purification we detected only minor amounts of copepod-released metabolites compared to background signals (Fig. 1). This is typical for plankton studies, where low abundance signals are encountered in a complex matrix (*Barofsky, Vidoudez & Pohnert, 2009*; *Vidoudez & Pohnert, 2012*). For organisms that tolerate crowding, it is possible to improve the signal to noise ratio by reducing the incubation volume e.g., by keeping organisms in the SPE reservoir. Weber and colleagues *(2013)* found that isotope labelling of autotrophic organisms enhance the possibility to filter out low concentration compounds from the labeled organism from complex chemical backgrounds. This could in principle be done with copepods too, but the labeling procedure requires copepods reared on isotope enriched food.

The most probable elemental composition was calculated for each metabolite, based on the accurate mass and isotope distribution pattern. Full structure elucidation requires more experiments and often more materials. Here we only identified the most obvious peak in the chromatogram; compound 10 (Table 2) as copepodamide G or an isomer of copepodamide G. The copepodamides are known to induce toxin production in harmful algal bloom-forming dinoflagellates. Their presence here demonstrates that the method is indeed sensitive enough to sequester and analyse infochemicals. Copepodamide G was among the ten compounds of highest relative abundance in both sexes which further indicates that copepodamides may be among the leading features of the copepod exometabolome. *Temora* produce three other copepodamides (C Berglund, pers. comm, 2015). All three share molecular scaffold with copepodamide G. In fact, copepodamide G is the lysolipid of the other three, and it is likely that the lengthy incubations without food resulted in an overrepresentation of copepodamide G due to degradation of the other copepodamides. Coepodamide G is substantially more water soluble than the other

copepodamides in *Temora*, which may also have contributed to an overrepresentation of copepodamide G compared to other copepodamides.

The hypothesized difference between male and female exudates was present but subtle. Only two out of the 87 compounds were exclusively found in female exudates (Table 2). Most compounds were, however, more abundant in female exudates, possibly reflecting higher metabolic turnover in the larger and egg-producing females. The high level of similarity between male and female exudates may contribute to the low specificity described for mate finding behavior in *Temora* and other copepods, where males and heterospecifics are sometimes followed by males (*Goetze, 2008*; *Goetze & Kiørboe, 2008*).

Female exudates did not trigger mate search behavior in male *T. longicornis*, which suggests that other or additional compounds or hydromechanical signatures are needed to trigger mate attraction. A lack of response could result if pheromones were not retained, or did not survive sample preparation. It could also result if the *Temora* females were recently mated and therefore not emitted pheromones (*Heuschele & Kiørboe, 2012*), or if males were not responsive. Thus, it is not possible to exclude that the metabolites listed here may be involved in mate finding in *T. longicornis*. Pilot bioassay experiments with both water conditioned by live females, and SPE extracts of female exudates obtained in a similar way generated trail following behavior in males, showing that the bioassay method is capable of indicating pheromones. Copepod pheromones can be predicted to be of low molecular weight as smaller molecules form longer trails per unit mass exuded (*Bagoien & Kiørboe, 2005*). Aquatic info-chemicals, however, encompass diverse classes of molecules, and molecular masses ranging from a molecular weight below 100 Da to protein signals larger than 31.5 kDa (*Pohnert, Steinke & Tollrian, 2007*). Metabolomic approaches targeting unknown infochemicals should consequently aim to cover as much as possible of the exometabolome to be useful.

Chemical interactions in the aquatic realm sometimes supersede the direct trophic interactions between organisms (*Cyr & Pace, 1993*; *Hay, 2009*; *Preisser, Bolnick & Benard, 2005*). Deciphering the chemical interplay between plankton organisms is consequently key to further our understanding of the pelagic ecosystem. Furthermore, insights to chemical signaling opens up applied possibilities such as design of artificial baits, pest management, and aquaculture tools similar to developments in terrestrial chemical ecology.

In summary, the closed loop solid phase extraction and metabolomic profiling of exudates from living copepods did provide information on the composition of the copepod exometabolome. The high relative abundance of copepodamide G shows that the method can be used to target info-chemicals from small plankton organisms. There is a subtle difference in the composition of male and female exudates, but negative trail following assays indicates that the sex pheromone of *Temora longicornis* involved other or additional compounds. Future studies should strive to include more polar compounds, larger compounds, and volatiles to create a more comprehensive view of the copepod exometabolome.

## ACKNOWLEDGEMENTS

Two anonymous referees provided comments that helped us improve the manuscript.

### Funding

This work was supported by the Danish National Strategic Research Council—IMPAQ—grant no. 10-093522, the Swedish Research Council Formas contract no. 223-2012-693, and EU FP7 ASSEMBLE grant agreement no. 227799. GMN was supported by a Linnaeus grant from the Swedish Research Councils (FORMAS and VR), and the Olle Engkvist Byggmästare Foundation. The Centre for Ocean Life is supported by the Villum Foundation. The funders had no role in study design, data collection and analysis, decision to publish, or preparation of the manuscript.

### Grant Disclosures

The following grant information was disclosed by the authors:
Danish National Strategic Research Council—IMPAQ: 10-093522.
The Swedish Research Council Formas: 223-2012-693,
EU FP7 ASSEMBLE: 227799.
Swedish Research Councils (FORMAS and VR).
Olle Engkvist Byggmästare Foundation.
Villum Foundation.

### Competing Interests

The authors declare there are no competing interests.

### Author Contributions

- Erik Selander conceived and designed the experiments, performed the experiments, analyzed the data, wrote the paper, prepared figures and/or tables, reviewed drafts of the paper.
- Jan Heuschele and Göran M. Nylund conceived and designed the experiments, performed the experiments, analyzed the data, contributed reagents/materials/analysis tools, wrote the paper, prepared figures and/or tables, reviewed drafts of the paper.
- Georg Pohnert, Peter Tiselius and Thomas Kiørboe conceived and designed the experiments, performed the experiments, wrote the paper, reviewed drafts of the paper.
- Henrik Pavia and Oda Bjærke conceived and designed the experiments, performed the experiments, reviewed drafts of the paper.
- Larisa A. Pender-Healy conceived and designed the experiments, performed the experiments, analyzed the data, reviewed drafts of the paper.

### Data Availability

The raw data produced for the current manuscript is enclosed in Supplemental Information 1.

## Supplemental Information

Supplemental information for this article can be found online at http://dx.doi.org/10.7717/peerj.1529#supplemental-information.

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
