# Peer review of "Solid phase extraction and metabolic profiling of exudates from living copepods"

_PeerJ, doi:10.7717/peerj.1529_

## Round 0.1 · original submission · Minor Revisions

Both submitted reviews suggest only minor changes to the manuscript and the third reviewer, who is struggling for time agrees. He will try and get some comments to us early next week. I have suggested some minor amendments.

Reviewer 1 ·

Basic reporting

The authors describe a solid phase extraction method for isolating exuded metabolites from copepods. These bioactive compounds in the medium play key roles in mate finding and inducing defensive responses to predators and it is therefore important to understand their chemistry.

Experimental design

Line 50 – In this section, I would mention the contact chemicals used by Tigriopus males to discriminate conspecific females. Even though these are not dissolved in the medium, they are still gender-specific exudates and infochemicals.
Line 101 – Because diet could strongly affect copepod exudates, the authors should provide details about culturing the algae and the sources of these strains. Also, please state how long the copepods were cultured in lab before initiating experiments. Were females gravid?
Line 106 – Where and when was this batch of “purified seawater” obtained and how was it stored? A basic profile of the amount of DOC and its composition would be useful. How could others repeat this experiment without more details about the base water?
Line 120 – How long were the copepods incubated in the pSW? Clearly the duration of incubation is critical to how much exudate accumulates.
Line 121 – It is conceivable that this exceptionally high density may affect the rate of exudate excretion. Any comments on the magnitude of such an effect?

Validity of the findings

Line 164 – Simply because a sample deviates from other replicate samples is a weak reason for excluding the data, especially when there are only 5 replicates. Did this exclusion reduce sample size or was another sample analyzed?
Line 239-43 – Is there any way to relate these concentrations to pheromone concentrations in nature? Otherwise, how do you know that copepod responses to the high lab concentrations of pheromones are not simply a lab artifact?

Reviewer 2 ·

Basic reporting

No Comments

Experimental design

No problems with the design

Validity of the findings

No comments

Additional comments

Review manuscript #2015:09:6829:0:0:REVIEW, entitled "Solid phase extraction and metabolic profiling of exudates from living copepods." By Selander et al.

The authors present a study on chemical exudates released by the marine copepod Temora longicornis. Copepods have been shown to produce and release a range of chemicals which are used as sex pheromones or as kairomones by their prey. The authors try to analyze the exometabolome, the released chemicals, and use metabolic profiling to identify exudate differences between sexes. They developed a closed loop extraction system with solid phase cartridges to isolate the chemicals from water and they developed a bioassay to identify responses related to the detection of sex pheromones. The authors detected 87 chemicals which were present in the copepod medium but not in the control water. Seven compounds were concentrated significantly higher in water holding females and two chemicals were exclusively present in the female treatment. The trail following bioassay showed no significant effects.
The analyses are convincing, although I am not the analytical expert, writing is generally concise and statistics are adequate. I have only minor comments.

Comments:
1. The authors discuss the limitations of the solid phase approach. Not all compound classes can be captured. Is it not to expect that sex pheromones are at least partly volatile? In this case they should create a recent “short term” track instead of a landscape of different tracks from the past.
2. L. 22: maybe replace “and“ by “or“ avoiding the impression that the same compounds will mediate both.
3. L. 43: is there any evidence that copepods change the chemical composition of the surrounding water by taking up chemical compounds? Related to cues, there will be much higher concentrations and depletions of infochemicals will not be likely?
4. L. 48 - Intro: I actually like some historical overview. Old papers showing the presence of sex pheromones in copepods are available, e.g., EVIDENCE FOR SEX PHEROMONES IN PLANKTONIC COPEPODS
by Steven Katona. L&O 1973.
5. L. 61: Is it really unforeseen? In most cases defenses will be induced leading to some reduced mortality?
6. L. 84: better add that this reduces mortality, otherwise the reader cannot interpret this information.
7. L. 110: Is it possible to add some data to the supplements to provide an idea about the magnitude of the effect?
8. L. 139: better “ultrapure water” with the company in brackets.
9. L. 148: how often were the copepods reused? The information is necessary to avoid the impression of pseudo-replication here.
10. L. 163: Is there any idea why two chromatograms deviated?
11. L. 204: How were the “noise peaks” defined?
12. L. 242: Please provide information about the volume of the copepod incubation, otherwise the concentration cannot be estimated.
13. L. 282: No mate tracking behavior was observed. Does this mean no trail following at all or no significant difference?
14. L. 283: this sentence is an interpretation and might go into the discussion.
15. L. 331: It is true that you likely found copepodamide G, but what about the other seven copepodamids. Why have they not been detected? This is an important question when you intend to prove that the system is sensitive enough to analyze infochemicals.
16. L. 366: I would not call it “detailed”
17. L. 369: I would provide a bit more discussion of the bioassay. Have you any prove that the bioassay functions at all? Maybe the trail was not recognized as such? You explained why you used a relatively high concentration of copepod cues, but have you tested lower ones? Yen et al. (J. Plankton Res. (2011) doi: 10.1093/plankt/fbq164) for example used relatively low cue concentrations in their freshwater copepod. Is it possible that high cue concentrations may prevent directional information?
18: Fig. 5: If the structure is not from this study I think it is not necessary here?


End of review

---

## Round 0.2 · accepted · Accept

Thank you for responding so positively to the review comments.